# Colossal barocaloric effects near room temperature in plastic crystals of neopentylglycol

P. Lloveras [1], A. Aznar[1], M. Barrio [1], Ph. Negrier[2], C. Popescu [3], A. Planes[4], L. Mañosa [4], E. Stern-Taulats[5], A. Avramenko[5], N.D. Mathur [5], X. Moya [5] & J.-Ll. Tamarit [1]

There is currently great interest in replacing the harmful volatile hydrofluorocarbon fluids used in refrigeration and air-conditioning with solid materials that display magnetocaloric, electrocaloric or mechanocaloric effects. However, the field-driven thermal changes in all of these caloric materials fall short with respect to their fluid counterparts. Here we show that plastic crystals of neopentylglycol $(CH_3)_2C(CH_2OH)_2$ display extremely large pressure-driven thermal changes near room temperature due to molecular reconfiguration, that these changes outperform those observed in any type of caloric material, and that these changes are comparable with those exploited commercially in hydrofluorocarbons. Our discovery of colossal barocaloric effects in a plastic crystal should bring barocaloric materials to the forefront of research and development in order to achieve safe environmentally friendly cooling without compromising performance.

[1] Grup de Caracterizació de Materials, Departament de Física, EEBE and Barcelona Research Center in Multiscale Science and Engineering, Universitat Politècnica de Catalunya, Eduard Maristany, 10-14, 08019 Barcelona, Catalonia, Spain. [2] Université de Bordeaux, LOMA, UMR 5798, F-33400 Talence, France. [3] CELLS-ALBA Synchrotron, E-08290 Cerdanyola del Vallès, Barcelona, Catalonia, Spain. [4] Departament de Física de la Matèria Condensada, Facultat de Física, Universitat de Barcelona, Martí i Franquès 1, 08028 Barcelona, Catalonia, Spain. [5] Department of Materials Science, University of Cambridge, Cambridge CB3 0FS, UK. Correspondence and requests for materials should be addressed to X.M. (email: xm212@cam.ac.uk) or to J.-L.T. (email: josep.lluis.tamarit@upc.edu)

Plastic crystals (PCs), also known as orientationally disordered crystals, are materials that lie at the boundary between solids and liquids[1]. They are normally made of nearly spherical small organic molecules whose centres of mass form a regular crystalline lattice[1], unlike liquid crystals that normally comprise highly anisotropic organic molecules with no long-range positional order[2]. The globular shape of these molecules provides little steric hindrance for reorientational processes, such that plastic crystals tend to be highly orientationally disordered away from low temperature[3]. This dynamical disordering often implies high plasticity under uniaxial stress, and hence the materials are known as plastic crystals[4]. On cooling, plastic crystals typically transform into ordered crystals (OCs) of lower volume via first-order phase transitions, whose latent heats arise primarily due to thermally driven large changes of orientational order, and this has led to proposals for passive thermal storage[5,6]. Here we exploit commercially available samples of the prototypical plastic crystal neopentylglycol (NPG), i.e., 2,2-dymethyl-1,3-propanediol. This material is an alcoholic derivative of neopentane $C(CH_3)_4$ made from cheap abundant elements, and enjoys widespread use in industry as an additive in the synthesis of paints, lubricants and cosmetics.

We achieve colossal pressure-driven thermal changes (barocaloric effects) near room temperature that are an order of magnitude better than those observed in state-of-the-art barocaloric (BC) materials[7–17] and comparable to those observed in the standard commercial hydrofluorocarbon refrigerant R134a[18] (Table 1). Our BC effects are colossal because the first-order PC-OC transition displays an enormous latent heat that is accompanied by an enormous change in volume, such that moderate applied pressure is sufficient to yield colossal thermal changes via the reconfiguration of globular neopentylglycol molecules (whose steric hindrance is low[3]). Moreover, reversibility is achieved at temperatures above the hysteretic transition regime. Our higher operating pressures do not represent a barrier for applications because they can be generated by a small load in a large volume of material via a pressure-transmitting medium, e.g., using a vessel with a neck containing a driving piston, whose small area is compensated by its distance of travel. Therefore, our demonstration of colossal BC effects in commercially available plastic crystals should immediately open avenues for the development of safe and environmentally friendly solid-state refrigerants.

## Results

### PC-OC phase transition in NPG at atmospheric pressure.
At room temperature and atmospheric pressure, NPG adopts an ordered monoclinic structure ($P2_1/c$) with four molecules per unit cell[19] (Fig. 1a). On heating, the material undergoes a reversible structural phase transition to a cubic structure ($Fm\bar{3}m$) with four molecules per unit cell that adopt an orientationally disordered configuration at any typical instant[20] (Fig. 1a). The first-order structural phase transition yields sharp peaks in $dQ/|dT|$ ($Q$ is heat, $T$ is temperature) recorded on heating and cooling (Fig. 1a), with a well-defined transition start temperature $T_0 \sim 314$ K on heating (Supplementary Fig. 1). By contrast, as a consequence of the nominally isothermal character of the PC-OC transition[21], the temperature ramp rate influences the transition finish temperature on heating, and the transition start and finish temperatures on cooling (e.g., by up to ~5 K for 1–10 K min$^{-1}$, Supplementary Fig. 1). Integration of the calorimetric peaks yields a large latent heat of $|Q_0| = 121 \pm 2$ kJ kg$^{-1}$ on heating, and $|Q_0| = 110 \pm 2$ kJ kg$^{-1}$ on cooling (Fig. 1a). These values of $|Q_0|$ are independent of the temperature ramp rate (Supplementary Fig. 1), and in good agreement with previous experimental values[1,22,23] of $|Q_0| \sim 123$–131 kJ kg$^{-1}$.

Integration of $(dQ/|dT|)/T$ and $C_p/T$ (Fig. 1b), permits the evaluation of entropy $S'(T) = S(T) - S(250$ K$)$ over a wide temperature range (Fig. 1c), as explained in the Experimental Section ($C_p$ is specific heat at atmospheric pressure). The large entropy change at the transition ($|\Delta S_0| \sim 383$ J K$^{-1}$ kg$^{-1}$ on heating and $|\Delta S_0| \sim 361$ J K$^{-1}$ kg$^{-1}$ on cooling) is in good agreement with previous experimental values[1,21–23] of $|\Delta S_0| \sim 390$–413 J K$^{-1}$ kg$^{-1}$. This large value of $|\Delta S_0|$ arises due to a non-isochoric order-disorder transition in molecular configurations, such that it exceeds values of $|\Delta S_0| << 100$ J K$^{-1}$ kg$^{-1}$ for first-order structural phase transitions associated with changes of ionic position[24–27] and electronic densities of states[24,27,28]. Consequently, the configurational degrees of freedom that are accessed via the non-isochoric order-disorder transition in our solid material yield entropy changes that compare favourably with those associated with the translational degrees of freedom accessed via solid-liquid-gas transitions in various materials[29], including the hydrocarbon fluids used for commercial refrigeration[18].

**Table 1 Barocaloric effects near first-order phase transitions**

| Compound | $T$ [K] | $|\Delta S|$ [J K$^{-1}$ kg$^{-1}$] | $|p|$ [GPa] | Reversible | Ref. |
|---|---|---|---|---|---|
| NPG | 320 | 445 | 0.25 | Yes | This work |
| NPG | 320 | 500 | 0.52 | Yes | This work |
| Ni$_{49.26}$Mn$_{36.08}$In$_{14.66}$ | 293 | 24 | 0.26 | partially | 7 |
| LaFe$_{11.35}$O$_{0.47}$Si$_{1.2}$ | 237 | 8.6 | 0.20 | partially | 8 |
| Gd$_5$Si$_2$Ge$_2$ | 270 | 11 | 0.20 | partially | 9 |
| Fe$_{49}$Rh$_{51}$ | 308 | 12 | 0.25 | partially | 10 |
| Mn$_3$GaN | 285 | 22 | 0.14 | partially | 11 |
| (MnNiSi)$_{0.62}$(FeCoGe)$_{0.38}$ | 330 | 70 | 0.27 | yes | 12 |
| BaTiO$_3$ | 400 | 1.6 | 0.10 | yes | 13 |
| (NH$_4$)$_2$SO$_4$ | 219 | 60 | 0.10 | yes | 14 |
| (NH$_4$)$_2$SnF$_6$ | 105 | 61 | 0.10 | yes | 15 |
| [TPrA]Mn[dca]$_3$ | 330 | 30 | 0.007 | yes | 16 |
| AgI | 390 | 60 | 0.25 | yes | 17 |
| Fluid R134a | 310 | 520 | 0.001 | yes | 18 |

Isothermal entropy change $|\Delta S|$ at temperature $T$ due to changes of hydrostatic pressure $|p|$ (the nearby values of transition temperature $T_0$ appear in Supplementary Table 1). All entries for barocaloric solids denote data derived from quasi-direct measurements[30]. For the fluid hydrofluorocarbon R134a (1,1,1,2-tetrafluoroethane, i.e. CH$_2$FCF$_3$), the value of $|\Delta S|$ represents the full condensation of the fluid at 310 K and 0.001 MPa, when exploited in a typical vapour-compression refrigeration cycle[18]

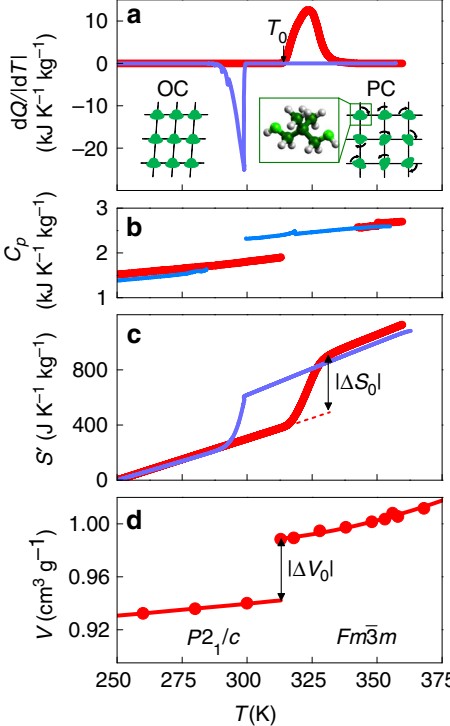

**Fig. 1** Thermally driven phase transition in NPG at atmospheric pressure. **a** Measurements of $dQ/|dT|$ after baseline subtraction, on heating (red) and cooling (blue) across the first-order cubic-monoclinic phase transition, revealing a large latent heat. The insets represent simplified plan views of the globular $(CH_3)_2C(CH_2OH)_2$ molecules (C = dark green spheres, H = grey spheres and O = light green spheres), which are configurationally ordered in the monoclinic ordered-crystal (OC) phase (left inset), and configurationally disordered in the cubic plastic-crystal (PC) phase (right inset). We assume only one molecule per unit cell for ease of representation. **b** Specific heat $C_p$ either side of the transition on heating (red) and cooling (blue). **c** Entropy $S'(T) = S(T) - S(250 \text{ K})$, evaluated via $S'(T) = S(T) - S(250 \text{ K}) = \int_{250 \text{ K}}^{T} (C_p + |dQ/dT|)/T'dT'$, revealing a large entropy change $|\Delta S_0|$ for the transition. **d** Specific volume $V(T)$ on heating, revealing a large volume change $|\Delta V_0|$ for the transition. Symbols represent experimental data, lines are guides to the eye.

On heating through the transition, x-ray diffraction data confirm the expected changes in crystal structure[19,20] (Supplementary Figs. 6 and 7). The resulting specific volume $V$ undergoes a large ~4.9% increase of $\Delta V_0 = 0.046 \pm 0.001 \text{ cm}^3 \text{g}^{-1}$ across the transition, for which $(\partial V/\partial T)_{p=0} > 0$ (Fig. 1d), presaging large conventional BC effects that may be evaluated[30] by using the Maxwell relation $(\partial V/\partial T)_p = -(\partial S/\partial p)_T$ to calculate the isothermal entropy change $\Delta S(p_1 \rightarrow p_2) = -\int_{p_1}^{p_2} (\partial V/\partial T)_p dp$ due to a change in pressure from $p_1$ to $p_2$. Near the transition, the volumetric thermal expansion coefficients for the OC and the PC phases are both $\sim 10^{-4} \text{ K}^{-1}$, implying the existence of additional[15] BC effects $\Delta S_+$ that are large and conventional at temperatures lying on either side of the transition. These additional BC effects are evaluated here using the aforementioned Maxwell relation, for changes in pressure $|p - p_{atm}| \sim |p|$ where atmospheric pressure $p_{atm} \sim 0 \text{ GPa}$, to obtain $\Delta S_+(p) = -[(\partial V/\partial T)_{p=0}]p$, where $(\partial V/\partial T)_p$ is assumed to be independent of pressure[13,15,17] (Supplementary Fig. 4 shows the error in $(\partial V/\partial T)_p$ to be ~20% for the PC phase, which implies an error in the total entropy change $\Delta S$ of ~3%).

Two contributions to $|\Delta S_0|$ may be identified as follows. One is the configurational entropy[31,32] given by $M^{-1}R\ln \Omega$, where

$M = 104.148 \text{ g mol}^{-1}$ is molar mass, $R$ is the universal gas constant, and $\Omega$ is the ratio between the number of configurations in the PC and the OC phases. The other is the volumetric entropy[31,32] $(\bar{\alpha}/\bar{\kappa}) \Delta V_0$, where the coefficient of isobaric thermal expansion $\bar{\alpha}$ (Supplementary Fig. 4), and the isothermal compressibility $\bar{\kappa}$ (Supplementary Fig. 5), have both been averaged across the PC-OC transition. Molecules of $(CH_3)_2C(CH_2OH)_2$ display achiral tetrahedral symmetry[33] (point group $T_d$, subgroup $C_{3v}$), yielding one configuration in the OC phase and 60 configurations in the PC phase (10 molecular orientations that each possesses six possible hydroxymethyl conformations). Therefore the configurational entropy is $M^{-1}R\ln 60 \sim 330 \text{ J K}^{-1} \text{ kg}^{-1}$, and the volumetric entropy is $\sim 60 \text{ J K}^{-1} \text{ kg}^{-1}$ (data from Fig. 1d and Supplementary Fig. 3a). The resulting prediction of $|\Delta S_0| \sim 390 \text{ J K}^{-1} \text{ kg}^{-1}$ agrees well with the experimental values reported above, and the previously measured experimental values[1,21–23] reported above.

**PC-OC phase transition in NPG under applied pressure.** Measurements of $dQ/|dT|$ under applied pressure (Fig. 2a, b) reveal that the observed transition temperatures vary strongly with pressure (Fig. 2c), with $dT/dp = 113 \pm 5 \text{ K GPa}^{-1}$ for the start temperature on heating, and $dT/dp = 93 \pm 18 \text{ K GPa}^{-1}$ for the start temperature on cooling, for pressures $p < 0.1$ GPa (black lines, Fig. 2c). These values of $dT/dp$ are amongst the largest observed for BC materials (Supplementary Table 1), and indicate that the first-order PC-OC transition of width ~10 K (Fig. 2a, b) could be fully driven in either direction using $|\Delta p| \sim |p| \sim 0.1$ GPa. At higher pressures, values of $dT/dp$ fall slightly, but remain large (Fig. 2c).

Integration of $(dQ/|dT|)/T$ at finite pressure reveals that the entropy change $|\Delta S_0|$ decreases slightly with increasing pressure (Fig. 2d). This decrease arises because the additional entropy change $\Delta S_+(p)$ increases in magnitude on increasing temperature in the PC phase [$(\partial V/\partial T)_{p=0}$ at 370 K is ~240% larger than $(\partial V/\partial T)_{p=0}$ at 320 K, Fig. 1d], whereas it is nominally independent of temperature in the OC phase near the transition. The fall seen in both $dT/dp$ and $|\Delta S_0|$ implies via the Clausius–Clapeyron equation $dT/dp = \Delta V_0/\Delta S_0$ that there is a reduction in $|\Delta V_0|$ at finite pressure (Fig. 2e), as confirmed using pressure-dependent dilatometry (Supplementary Fig. 3a) and pressure-dependent x-ray diffraction (Supplementary Fig. 3b).

In order to plot $\Delta S(T,p)$, we obtained finite-pressure plots of $S'(T,p) = S(T,p) - S(250 \text{ K},0)$ (Fig. 3a, b) by integrating the data in Fig. 2a, b and Fig. 1b, and displacing each corresponding plot by $\Delta S_+(p)$ at 250 K, as explained in the Experimental Section. (Note that $\Delta S_+(p)$ was evaluated below $T_0(p=0)$ to avoid the forbidden possibility of $T_0(p)$ rising to the temperature at which $\Delta S_+(p)$ was evaluated at high pressure.) From Fig. 3a, b, we see that the entropy change associated with the transition $\Delta S_0(p)$ combines with the smaller same-sign additional entropy change $\Delta S_+(p)$ away from the transition, yielding total entropy change $\Delta S(p)$.

**BC performance.** By following isothermal trajectories in our plots of $S'(T,p)$ obtained on cooling (Fig. 3b), we were able to evaluate $\Delta S(T,p)$ on applying pressure (Fig. 3c), as cooling and high pressure both tend to favour the low-temperature low-volume OC phase. Similarly, by following isothermal trajectories in our plots of $S'(T,p)$ obtained on heating (Fig. 3a), we were able to evaluate $\Delta S(T,p)$ on decreasing pressure (Fig. 4c), as heating and low pressure both tend to favour the high-temperature high-volume PC phase.

Discrepancies in the magnitude of $\Delta S(T,p)$ on applying and removing pressure (Fig. 3c) are absent in the range ~314-342 K, evidencing reversibility. Our largest reversible isothermal entropy

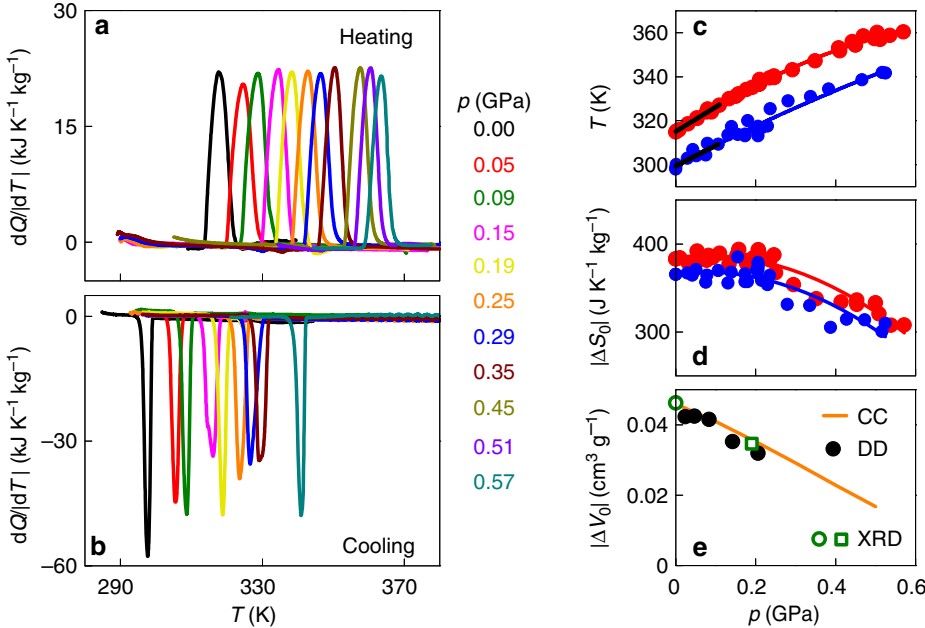

**Fig. 2** Pressure-driven phase transition in NPG. **a**, **b** Measurements of d$Q$/|d$T$| on heating and cooling across the first-order PC-OC transition for different values of increasing pressure $p$, after baseline subtraction. **c**, **d** Transition temperature and entropy change |$\Delta S_0(p)$| on heating (red symbols) and cooling (blue symbols), derived from the calorimetric data of **a**, **b** and equivalent data at other pressures (shown in Supplementary Fig. 2). Black lines in **c** are linear fits. Red and blue lines in **c**, **d** are guides to the eye. **e** Volume change for the transition |$\Delta V_0(p)$|: solid symbols obtained from the dilatometric data (DD) in Supplementary Fig. 3a; open circle obtained from the x-ray diffraction data in Fig. 1c, open square obtained from the x-ray diffraction data in Supplementary Fig. 3b; orange line obtained from **c**, **d** via the Clausius–Clapeyron (CC) equation

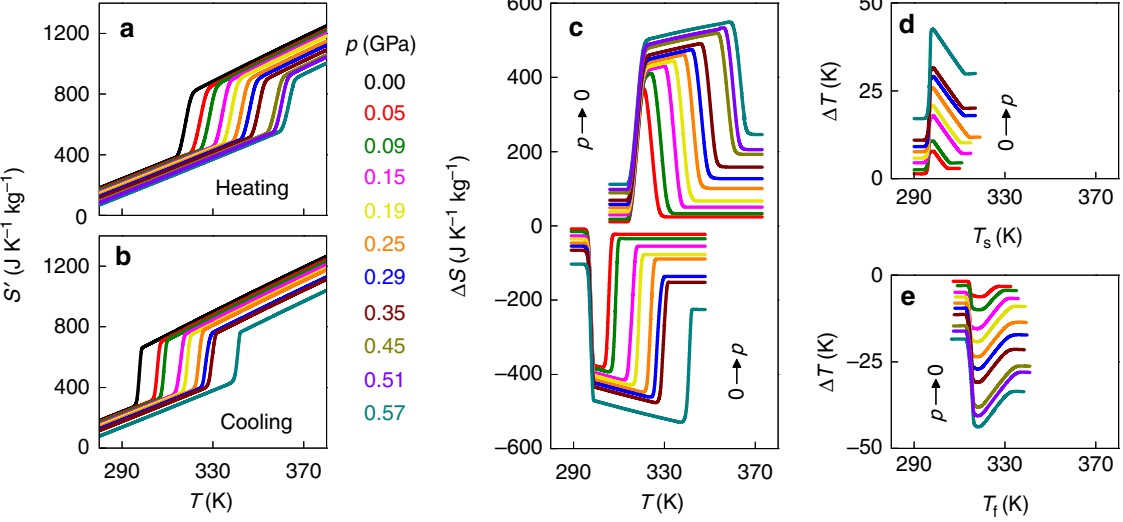

**Fig. 3** Colossal barocaloric effects in NPG near room temperature. **a**, **b** Entropy $S'(T,p)$ with respect to the absolute entropy at 250 K and $p \sim 0$, on **a** heating and **b** cooling through the first-order PC-OC phase transition. **c** Isothermal entropy change $\Delta S$ for $0 \rightarrow p$ deduced from **b**, and for $p \rightarrow 0$ deduced from **a**. **d** Adiabatic temperature change $\Delta T$ versus starting temperature $T_s$, for $0 \rightarrow p$ deduced from **b**. **e** Adiabatic temperature change $\Delta T$ versus finishing temperature $T_f$ for $p \rightarrow 0$ deduced from **a**

change |$\Delta S$| ~ 510 J K⁻¹ kg⁻¹ arises at ~330 K for |$p$| ~ 0.57 GPa, and substantially exceeds the BC effects of |$\Delta S$| ≤ 70 J K⁻¹ kg⁻¹ that were achieved using similar values of |$p$| in a range of materials near room temperature (Fig. 4a), namely magnetic alloys[7–12,34], ferroelectric[13,35,36] and ferrielectric[15] materials, fluorides and oxifluorides[14,37–40], hybrid perovskites[16], and superionic conductors[17,41,42]. Moreover, our largest value of |$\Delta S$| substantially exceeds the values recorded for magnetocaloric[30,43–46], electrocaloric[30,47,48], and elastocaloric[30,49]

materials, and is comparable to the values observed in the standard commercial hydrofluorocarbon refrigerant fluid R134a[18], for which |$\Delta S$| = 520 J K⁻¹ kg⁻¹ at ~310 K for much smaller operating pressures of ~0.001 GPa (Fig. 4a). We can also confirm that NPG compares favourably with other BC solids[7–12,31] when normalizing the peak entropy change by volume[30] to yield |$\Delta S$| ~ 0.54 J K⁻¹ cm⁻³ (the NPG density is 1064 kg m⁻³ at ~320 K). (While finalizing our manuscript, which is based on our 2016 patent, we learned about the pre-print of ref. [50], which lists

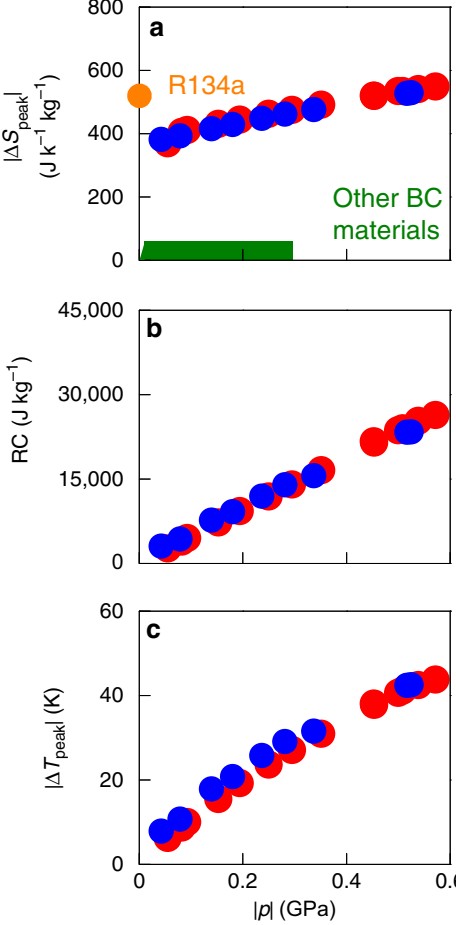

**Fig. 4** Barocaloric performance near room temperature. **a** For NPG, we show the peak isothermal entropy change $|\Delta S_{peak}|$ for pressure changes of magnitude $|p|$, on applying pressure (blue symbols) and removing pressure (red symbols). For comparison, the green envelope represents state-of-the-art barocaloric materials (Table 1) that operate near room temperature, and the orange symbol represents the standard commercial fluid refrigerant[18] R134a for which operating pressures are ~0.001 GPa. For NPG alone, we show the variation with $|p|$ of **b** refrigerant capacity RC = $|\Delta S_{peak}|$ × [FWHM of $\Delta S(T)$] and **c** peak values of the adiabatic temperature change $|\Delta T_{peak}|$, on applying pressure (blue symbols) and removing pressure (red symbols) near room temperature

literature values of thermally driven entropy changes for PC-OC transitions in NPG and other plastic crystals at atmospheric pressure, and suggests they could be used in barocalorics.)

The large variation of transition temperature with pressure (Fig. 2c) permits large entropy changes of $|\Delta S| \sim 445\,\mathrm{J\,K^{-1}\,kg^{-1}}$ to be driven with relatively moderate pressure changes of $|p| \sim 0.25\,\mathrm{GPa}$ (Fig. 3c), yielding giant BC strengths[30] of $|\Delta S|/|p| \sim 1780\,\mathrm{J\,K^{-1}\,kg^{-1}\,GPa^{-1}}$. Larger pressures extend the reversible BC effects to higher temperatures (Fig. 3c), causing the large refrigerant capacity RC to increase (Fig. 4b) despite the slight reduction in $|\Delta S_0(p)|$ (Fig. 2d). The BC effects in NPG are so large (Fig. 4a) that unpractical changes of pressure would be required to achieve comparable RC values in other BC materials.

By following adiabatic trajectories in $S'(T, p)$ (Fig. 3a, b), we established both the adiabatic temperature change $\Delta T(T_s, p)$ on applying pressure $p$ at starting temperature $T_s$ (Fig. 3d), and the adiabatic temperature change $\Delta T(T_f, p)$ on removing pressure $p$ to reach finishing temperature $T_f$ (Fig. 3e). On applying our

largest pressure ($p \sim 0.57$ GPa), an adiabatic temperature increase of $\Delta T \sim 30$ K with respect to $T_s \sim 318$ K is necessarily reversible above the thermally hysteretic regime, such that an equivalent temperature change of opposite sign is achieved on pressure removal. These BC effects substantially exceed both the BC effects of $|\Delta T| \leq 10$ K that were achieved in inorganic materials[7–10,12] by exploiting room-temperature phase transitions with similar values of $|p|$; and substantially exceeds the BC effects of $|\Delta T| \sim 9$ K that were achieved[51] away from a phase transition with a smaller value of $|p| = 0.18$ GPa in organic poly(methyl methacrylate) at $T_s \sim 368$ K.

## Discussion

To exploit our material in BC cooling devices, the non-monolithic working body and its intermixed pressure-transmitting medium may exchange heat with sinks and loads via fluid in a secondary circuit, heat pipes or fins[52]. The requisite high pressures could be generated in large volumes using small loads and small-area pistons, just as small voltages can generate large electric fields in the many thin films of an electrocaloric multilayer capacitor[53,54]. To improve the BC working body, it would be attractive to decrease the observed hysteresis using both chemical and physical approaches, enhance the limited thermal conductivity e.g., by two orders of magnitude via the introduction of graphite matrices[55], and combine different plastic crystals that operate at quite different temperatures[1,52,56]. More generally, our observation of colossal and reversible BC effects in NPG should inspire the study of BC effects in other mesophase systems that lie between liquids and solids, most immediately other organic plastic crystals whose PC-OC transitions display large latent heats and large volume changes[52].

After acceptance of our paper, ref.[50] by Li et al. was published in *Nature*. In the published version they reported a barocaloric entropy change of 389 J K⁻¹ kg⁻¹ for NPG. This value is lower than our value because these authors used lower pressure, and only considered the contribution from the PC-OC transition, while as shown in our manuscript the contributions beyond the transition are relevant for NPG and can be as large as ~80 J K⁻¹ kg⁻¹ for our ~0.25 GPa driving pressure.

## Methods

**Samples**. NPG of purity of 99% was purchased as a powder from Sigma-Aldrich. The typical grain size was ~100 µm, as determined using optical microscopy.

**Techniques**. Measurements of $dQ/|dT| = \frac{dQ/dt}{|dT/dt|}$ were performed at atmospheric pressure in a commercial TA Q100 differential scanning calorimeter (DSC), at $\pm 1$–10 K min⁻¹, using ~10–20 mg samples of NPG ($t$ is time).

Measurements of specific heat $C_p$ were performed at atmospheric pressure in a commercial TA Q2000 DSC, at $\pm 5$ K min⁻¹, using ~20 mg samples of NPG. Values of $C_p$ were obtained by recording heat flow out of/into the sample as a function of temperature, and comparing it with the heat flow out of/into a reference sapphire sample under the same conditions[57]. Latent heat $|Q_0| = \left| \int_{T_1}^{T_2} \frac{dQ}{dT} dT \right|$ across the PC-OC transition was obtained after subtracting baseline backgrounds, with start temperature $T_1$ freely chosen below (above) the transition on heating (cooling), and finish temperature $T_2$ freely chosen above (below) the transition on heating (cooling).

Measurements of $dQ/dT$ were performed at constant applied pressure using two bespoke differential thermal analysers (DTAs). For applied pressures of < 0.3 GPa, we used a Cu-Be Bridgman pressure cell with chromel-alumel thermocouples. For applied pressures of < 0.6 GPa, we used a model MV1-30 high-pressure cell (Institute of High Pressure Physics, Polish Academy of Science) with Peltier elements as thermal sensors. The temperature of both pressure cells was controlled using a circulating thermal bath (Lauda Proline RP 1290) that permitted the measurement temperature to be varied at ~±2 K min⁻¹ in 183–473 K. NPG samples of mass ~100 mg were mixed with an inert perfluorinated liquid (Galden, Bioblock Scientist) to remove any residual air, and hermetically encapsulated inside Sn containers. The pressure-transmitting medium was DW-Therm (Huber Kältemaschinenbau GmbH). Entropy change $|\Delta S_0(p)| = \left| \int_{T_1}^{T_2} (dQ/dT)/T \, dT \right|$

across the PC-OC transition was obtained after subtracting baseline backgrounds, and the choice of $T_1$ and $T_2$ is explained above.

Variable-temperature high-resolution x-ray diffraction was performed at atmospheric pressure in transmission, using Cu K$\alpha_1$ = 1.5406 Å radiation in a horizontally mounted INEL diffractometer with a quartz monochromator, a cylindrical position-sensitive detector (CPS-120) and the Debye-Scherrer geometry. NPG samples were introduced into a 0.5-mm-diameter Lindemann capillary to minimize absorption, and the temperature was varied using a 600 series Oxford Cryostream Cooler. Using the Materials Studio software[58], lattice parameters were determined by pattern matching using the Pawley method for the cubic phase, and by Rietveld refinement for the monoclinic phase.

Dilatometry was performed using a bespoke apparatus that operated up to 0.3 GPa over a temperature range of ~193–433 K. Molten NPG samples of mass ~1 g were encapsulated inside stainless-steel containers to remove any residual air. Each container was then perforated by a stainless-steel piston, whose relative displacement with respect to a surrounding coil could be detected via measurement of electromotive force[59].

Variable-pressure x-ray diffraction measurements were performed at beamline MSPD BL04 in the ALBA-CELLS synchrotron[60], using an x-ray wavelength of 0.534 Å obtained at the Rh K-edge. The beamline is equipped with Kirkpatrick-Baez mirrors to focus the x-ray beam to 20 μm × 20 μm, and uses a Rayonix CCD detector. The NPG sample was placed with two small ruby chips at the centre of a 300 μm-diameter hole in a stainless steel gasket, preindented to a thickness of 55 μm. For room-temperature measurements, we used symmetric diamond-anvil cells (DACs) with diamonds of 700 μm. For high-temperature measurements, we used a gas-membrane driven DAC equipped with diamonds possessing 400 μm culets, and varied the temperature using a resistive heater. Temperature was measured using a K-type thermocouple attached to one diamond anvil, close to the gasket. The thermocouple was accurate to 0.4% in our measurement-set temperature range. For all the measurements, NaCl powder was used as the pressure marker[61]. The accuracy of pressure readings was ~±0.05 GPa. Indexing and refinement of the powder patterns were performed using the Materials Studio software, by pattern matching using the Pawley method.

**Construction of entropy curves**. Using specific heat data at atmospheric pressure (Fig. 1b), specific volume data at atmospheric pressure (Fig. 1d), and $dQ/|dT|$ data at constant pressure (Figs. 1a and 2a, b), we calculated $S'(T,p) = S(T,p) - S$ (250 K,0) using Eq. (1):

$$S'(T,p) = \begin{cases} \int_{250\text{K}}^{T} \frac{C_{OC}(T')}{T'} dT' + \Delta S_+(p) & T \leq T_1 \\ S(T_1,p) + \int_{T_1}^{T} \frac{1}{T'}\left(C_{OC-PC}(T') + \left|\frac{dQ(T',p)}{dT'}\right|\right) dT' + \Delta S_+(p) & T_1 \leq T \leq T_2 \\ S(T_2,p) + \int_{T_2}^{T} \frac{C_{PC}(T')}{T'} dT' + \Delta S_+(p) & T \geq T_2 \end{cases}$$

(1)

where $T_1$ is the transition start temperature, $T_2$ is the transition finish temperature, $C_{OC}$ is the specific heat of the OC phase, $C_{PC}$ is the specific heat of the PC phase, and $C_{OC-PC} = (1-x)C_{OC} + xC_{PC}$ represents the specific heat inside the transition region, where the transformed fraction $x$ on crossing the PC-OC transition was calculated using Eq. (2):

$$x = \left[\int_{T_1}^{T} (dQ/dT') dT'\right] / \left[\int_{T_1}^{T_2} (dQ/dT) dT\right]$$

(2)

All values of specific heat are assumed to be independent of pressure.

## Data availability

All relevant data are presented via this publication and Supplementary Information.

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

## Acknowledgements

This work was supported by the MINECO projects MAT2016-75823-R and FIS2017-82625-P, the DGU project 2017SGR-42, the UK EPSRC grant EP/M003752/1, and the ERC Starting grant no. 680032. We acknowledge ALBA for time on MSPD BL04 under proposal 2016021701. E.S.-T. and X.M. are grateful for support from the Royal Society.

## Author contributions

J.L.T., L.M. and X.M. conceived the study. J.L.T., M.B., P.L. and X.M. planned the research. A.Av. performed the calorimetric measurements at atmospheric pressure. M.B. performed the dilatometry measurements and the in-lab x-ray diffraction measurements. P.L. performed the calorimetric measurements under pressure. P.L., A.Az., E.S.-T. and X.M. performed the synchrotron x-ray diffraction measurements, with support from C.P. P.N. performed the analysis of the synchrotron x-ray data. Results were discussed by J.L.T., P.L., L.M., A.P., N.D.M. and X.M. X.M. wrote the manuscript with N.D.M. and P.L. using substantive feedback from J.L.T., L.M. and A.P. The remaining authors also contributed to the preparation of the manuscript.

## Additional information

**Competing interests:** The use of NPG and other plastic crystals for barocaloric cooling is covered in the following patent: X.M., A.Av., L.M., J-Ll.T. and P.L., Use of barocaloric materials and barocaloric devices, PCT/EP2017/076203 (2017). The remaining authors declare no competing interests.

