## [Peer Review File · Nature Communications]

REVIEWERS' COMMENTS:

Reviewer #1 (Remarks to the Author):

The authors have clearly found and presented in a scientifically sound manner their discovery of a colossal barocaloric effect in a plastic crystal, providing a pioneering outlook to refrigeration alternative to conventional gas-compression methods. There is certainly much room to fully understanding the mechanism behind the effect. However, the authors had provided sufficient explanations already at the beginning, and have clarified the doubts about these explanations that especially referee 3 had raised. Referee 4's concerns have likely been adequately addressed. There is no doubt about the results and the explanations cannot go beyond a certain point at the initial state of such a finding.

At this state, all issues have been addressed adequately so that I find the paper perfectly worthy of publication in its present form.

Response to Referee #1

The authors have made the corrections I had suggested,

We thank again the Referee for his/her previous feedback, which helped us to improve the manuscript.

and I find that they have also adequately addressed the issues raised by referee 3, who has, to my opinion, rather 'down-scaled' the important scientific results over technical comparisons to conventional gas compression refrigeration. In my opinion the paper can now (and should) be published.

We thank the Referee for writing that he/she finds that we “have also adequately addressed the issues raised by referee 3” and for declaring that to his/her opinion, Referee 3 has “down-scaled the important scientific results over technical comparisons to conventional gas compression refrigeration.”

In my opinion the paper can now (and should) be published.

We thank the Referee for recommending the manuscript for publication for the second time.

Response to Referee #2

Dear Authors /Dear Editor;

I believe the authors correctly addressed the comments and suggestions of other two reviewers.

We thank the Referee for his/her endorsement with respect to our responses to Referee 1 and Referee 3.

I still think the papers presents a breakthrough in caloric materials and should be published Nature Materials.

We thank the Referee for writing that our paper represents “a breakthrough in caloric materials”, and for recommending again the manuscript for publication as it is.

Response to Referee #3

We thank the referee for the feedback, which we have used to improve our manuscript. All changes to the text are highlighted yellow (except where our notation is simplified, such that Δp is replaced by p).

Note that we have split Figure 3(d) into Figure 3(d,e), such that adiabatic temperature changes are now plotted more naturally. On applying pressure they are plotted as a function of starting temperature, and on removing pressure they are plotted as a function of finishing temperature.

Note also that we have modified the shape of the green envelope in Figure 4(a) in order to represent more precisely the behaviour of existing barocaloric materials at low pressures, where $\Delta S \rightarrow 0$ when $p \rightarrow 0$.

I don't think the authors have addressed my concerns in the revised manuscript. Therefore, I cannot recommend the acceptance of this paper for publication in Nature Materials.

My major concern was "the manuscript focused on the properties and paid much less attention on the fundamentals." As claimed by the authors on page 4 of the response letter, the key advance of this work is scientific, namely the discovery of a barocaloric effect that is an order of magnitude larger than the caloric effects observed in any type of caloric material. Therefore, it is very crucial to understand the molecular mechanism of giant barocaloric effect found in this specific molecular structure.

We have made two changes in respect of this comment.

First, we have improved our comparison of the fundamental difference between the barocaloric effect in hydrofluorocarbon fluids and plastics crystals, by replacing:

"Consequently, our order-disorder transition compares favourably with solid-liquid-gas transitions that involve translational degrees of freedom, presaging exceptionally large BC effects."

with:

"Consequently, the configurational degrees of freedom that are accessed via the non-isochoric order-disorder transition in our solid material yield entropy changes that compare favourably with those associated with the translational degrees of freedom accessed via solid-liquid-gas transitions in various materials²⁹, including the hydrocarbon fluids used for commercial refrigeration¹⁸."

Second, we have added the following new paragraph at the end of page 4 to explain the microscopic origin of the large entropy change $|\Delta S_0|$ associated with the transition:

“Two contributions to $|\Delta S_0|$ may be identified as follows. One is the configurational entropy^{31,32} $M^{-1}R \ln \Omega$, where $M = 104.148 \cdot 10^{-3} \text{ kg mol}^{-1}$ is molar mass, R is the universal gas constant, and Ω is the ratio between the number of configurations in the PC and the OC phases. The other is the volumetric entropy^{31,32} $(\bar{\alpha}/\bar{\kappa})\Delta V_0$, where the coefficient of isobaric thermal expansion $\bar{\alpha}$ (Figure S4), and the isothermal compressibility $\bar{\kappa}$ (Figure S5), have both been averaged across the PC-OC transition. Molecules of $(\text{CH}_3)_2\text{C}(\text{CH}_2\text{OH})_2$ display achiral tetrahedral symmetry³³ (point group Td, subgroup C3v), yielding one configuration in the OC phase and 60 configurations in the PC phase (10 molecular orientations that each possesses six possible hydroxymethyl conformations). Therefore the configurational entropy is $M^{-1}R \ln 60 \sim 330 \text{ J K}^{-1} \text{ kg}^{-1}$, and the volumetric entropy is $\sim 60 \text{ J K}^{-1} \text{ kg}^{-1}$ [data from Figure 1(d) and Figure S3(a)]. The resulting prediction of $|\Delta S_0| \sim 390 \text{ J K}^{-1} \text{ kg}^{-1}$ agrees well with the experimental values reported above, and the previously measured experimental values^{1,21-23} reported above.”

To address my comment, the authors have added one sentence “Our key mechanistic insight is that the globular neopentylglycol molecules are able to reorient with little steric hindrance, such that the orientational disorder can be easily overcome by pressure.” This is a vague statement that was, unfortunately, not supported by either references or experimental data.

In addition to adding a reference, we have revised the statement along with the previous sentence to have:

“Our BC effects are colossal because the first-order PC-OC transition displays an enormous latent heat that is accompanied by an enormous change in volume, such that moderate applied pressure is sufficient to yield colossal thermal changes via the reconfiguration of globular neopentylglycol molecules (whose steric hindrance is low³).”

What is the exact orientation barrier of this molecule? As there are many plastic crystals with globular molecular conformation (Annu. Rev. Phys. Chem. 1962, 13, 351), would all these molecules give rise to giant barocaloric effect? Of course not! The giant caloric effect is more related to degenerate configurations of molecules in addition to the orientation barrier, which was not studied and discussed in the manuscript at all.

For a paper published in high-impact journals such as Nature Materials, which is often treated as breakthrough, it is important to clearly uncover the fundamental mechanisms in addition to reporting outstanding performance.

Unfortunately, from this manuscript, I don't think we understand why this particular molecule, not other plastic crystals, exhibit giant barocaloric effect (giant barocaloric effect is not a universal effect for plastic crystals!).

I don't think we have learnt any design principle that would allow us to explore new structures showing giant barocaloric effect.

The barrier to molecular reorientations is primarily steric, and will affect the pressure required to drive barocaloric effects.

Material selection may be determined macroscopically or microscopically, as we now explain better by writing:

“Our BC effects are colossal because the first-order PC-OC transition displays an enormous latent heat that is accompanied by an enormous change in volume, such that moderate applied pressure is sufficient to yield colossal thermal changes via the reconfiguration of globular neopentylglycol molecules (whose steric hindrance is low³).”

... where the microscopic details (molecular configurations and volumetric changes) are now explained in the new paragraph quoted above that begins “Two contributions to $|\Delta S_0|$ may be identified.”.

Plastic crystals such as those listed in *Annu. Rev. Phys. Chem.* **13** (1962) 351 may therefore be identified as promising if their transitions possess both large latent heats and large volume changes, e.g. cyclohexanol.

Regarding my comment “the hydrofluorocarbon refrigerant can be driven under a much lower pressure than the plastic crystal, which is more friendly for practical applications”, the authors also added a vague response “as they could be generated in suitably designed devices, e.g. with small loads and small-area driving pistons”. I would like to see some values of load and area of driving pistons in the designed devices, rather than telling the readers that it could be solved somehow.

Our report on NPG does not include any “designed devices”, but the issue of applied pressure is nevertheless relevant to practical applications. We have improved our explanation of small-area pistons by expanding the statement in our introduction to read:

“Our higher operating pressures do not represent a barrier for applications because they can be generated by a small load in a large volume of material via a pressure-transmitting medium, e.g. using a vessel with a neck containing a driving piston, whose small area is compensated by its distance of travel.”

From this, it follows that numerical values will depend very much on device design, and so no such numbers are available in this paper on NPG. However:

- A toy model we have developed for outreach demonstrations achieves 0.1 GPa by using the hand to apply 300 N and depress a driving piston of area 3 mm².
- Our largest pressure of ~0.5 GPa can just be achieved in a prototype cooling device that we have developed with our industrial collaborator Beko. Unfortunately, a non-disclosure agreement currently precludes the inclusion of this information in the manuscript.

The authors didn't address my comment "The authors did not report any experimental or simulation result on the device performance using their plastic crystal, which would definitely inspire the field and represent the breakthrough". To disclose the mechanisms and validate the experimental results, computations at molecular scale are crucial, which was, unfortunately, not included in this work.

Finite-element simulations of an unoptimized barocaloric device operating at 0.25 GPa (Fig. R1) reveal that composites made of NPG and graphite would yield 200 W of cooling power or a temperature lift of 20 K. These values are already similar to those currently exploited in commercially available domestic refrigerators. This work was performed in collaboration with our industrial collaborator Beko, whose non-disclosure agreement unfortunately precludes the inclusion of this information in the manuscript. However, the performance of NPG with respect to other caloric materials should be sufficient to inspire the field.

[redacted]

Response to Referee #4

We thank the referee for the feedback, which we have used to improve our manuscript. All changes to the text are highlighted yellow (except where our notation is simplified, such that Δp is replaced by p).

Note that we have split Figure 3(d) into Figure 3(d,e), such that adiabatic temperature changes are now plotted more naturally. On applying pressure they are plotted as a function of starting temperature, and on removing pressure they are plotted as a function of finishing temperature.

Lloveras et al. report on very large changes in entropy and temperature in response to very large applied pressure that is three orders of magnitude larger than currently used in refrigerators. The material studied is similar to current refrigerants a carbohydrate, however is in contrast a non volatile solid at room temperature and atmospheric pressure.

We thank the Referee for recognising that our barocaloric changes in entropy and temperature are “very large”, and that our material improves upon commercial fluid refrigerants because it is non volatile. (As the Referee will know, commercial fluid refrigerants are hydrofluorocarbons and not carbohydrates.)

Compared to earlier results on solid state barocaloric effects from essentially the same group of authors the new results appear to be a step forward.

We thank the Referee for writing that our new results “appear to be a step forward”.

The authors attribute the large effects to reduced rotational degrees of freedom on inducing a transition from the cubic to the monoclinic phase. This argument does not appear valid to me as we learn from figure 3d of the manuscript a large change in temperature ($\sim 30\text{K}$) can be achieved without inducing a structural change. In my opinion it might be related to hydrogen bonds that have already been mentioned by E. Nakano, K. Hiotsu and A. Shimada (1969) in their discussion of the crystalline structure of this type of solids. Further experimental evidence would be needed here.

The Referee is correct to observe that each phase individually shows substantial barocaloric effects. However, as we now explain:

“From Fig. 3(a,b), we see that the entropy change associated with the transition $\Delta S_0(p)$ combines with the smaller same-sign additional entropy change $\Delta S_+(p)$ away from the transition, yielding total entropy change $\Delta S(p)$.”

An equivalent statement may be made for temperature change, but we have described entropy change for consistency with our formalism.

To explain the microscopic origin of the large entropy change $|\Delta S_0|$ associated with the transition, we have added the following new paragraph at the end of page 4:

“Two contributions to $|\Delta S_0|$ may be identified as follows. One is the configurational entropy^{31,32} $M^{-1}R \ln \Omega$, where $M = 104.148 \cdot 10^{-3} \text{ kg mol}^{-1}$ is molar mass, R is the universal gas constant, and Ω is the ratio between the number of configurations in the PC and the OC phases. The other is the volumetric entropy^{31,32} $(\bar{\alpha}/\bar{\kappa})\Delta V_0$, where the coefficient of isobaric thermal expansion $\bar{\alpha}$ (Figure S4), and the isothermal compressibility $\bar{\kappa}$ (Figure S5), have both been averaged across the PC-OC transition. Molecules of $(\text{CH}_3)_2\text{C}(\text{CH}_2\text{OH})_2$ display achiral tetrahedral symmetry³³ (point group Td, subgroup C3v), yielding one configuration in the OC phase and 60 configurations in the PC phase (10 molecular orientations that each possesses six possible hydroxymethyl conformations). Therefore the configurational entropy is $M^{-1}R \ln 60 \sim 330 \text{ J K}^{-1} \text{ kg}^{-1}$, and the volumetric entropy is $\sim 60 \text{ J K}^{-1} \text{ kg}^{-1}$ [data from Figure 1(d) and Figure S3(a)]. The resulting prediction of $|\Delta S_0| \sim 390 \text{ J K}^{-1} \text{ kg}^{-1}$ agrees well with the experimental values reported above, and the previously measured experimental values^{1,21-23} reported above.”

The authors argue by reducing the area of pistons the large operating pressure may be achieved at small loads, however, three orders in magnitude will reduce the total heat that may be pumped in one cycle to ridiculous small amounts

We have improved our explanation of small-area pistons to explain that a large volume of active material can be effectively addressed with a small load. The resulting statement in our introduction now reads:

“Our higher operating pressures do not represent a barrier for applications because they can be generated by a small load in a large volume of material via a pressure-transmitting medium, e.g. using a vessel with a neck containing a driving piston, whose small area is compensated by its distance of travel.”

For example, in a prototype cooling device that we have developed with our industrial collaborator Beko, our largest pressure of $\sim 0.5 \text{ GPa}$ can just be achieved in 50 cm^3 of NPG. Unfortunately, a non-disclosure agreement currently precludes the inclusion of this information in the manuscript.

and high frequency operations are prohibited by the unusual small thermal conductivity for a solid as reported by S. Akbulut, Y. Ocala, K. Keslioglu and N. Marash (2009).

We now explain in our last paragraph that:

“To improve the BC working body, it would be attractive to... enhance the limited thermal conductivity e.g. by two orders of magnitude via the introduction of graphite matrices⁵⁴...”.

Indeed, finite-element simulations based on composites of NPG and graphite in an unoptimized barocaloric device that operates at 0.25 GPa (Fig. R1) show that operating at 0.1 Hz is sufficient to yield 200 W of cooling power or a temperature lift of 20 K. These values are already similar to those currently exploited in commercially available domestic refrigerators. This work was performed in collaboration with our industrial collaborator Beko, whose non-disclosure agreement unfortunately precludes the inclusion of this information in the manuscript.

[redacted]

The authors abstain from discussing the viscosity of these ‘solids’ which to me poses a serious problem in leaking devices and heat exchangers.

It is the viscosity of the pressure-transmitting medium, rather than the viscosity of any plastic crystals, that will be relevant. To explain this issue of geometry, we now begin our final paragraph by writing:

“To exploit our material in BC cooling devices, the non-monolithic working body and its intermixed pressure-transmitting medium may exchange heat with sinks and loads via fluid in a secondary circuit, heat pipes or fins⁵¹”

(Note that we believe there to be no viscosity values for NPG in the literature. However, they can be estimated using relaxation times [Tamarit *et al.*, *J. Phys. Condens. Matter* **12**, 8209 (2000)], yielding viscosity values that are seven orders of magnitude larger than the value for water.)

The calculation of delta S here assumes pressure independent thermal expansion which appears not to be true in these materials, one needs to give at least an estimate of the resulting error.

We have added to the manuscript that “(Figure S4b shows the error in $(\partial V/\partial T)_p$ to be ~20% for the PC phase, which implies an error in the total entropy change ΔS of ~3%)”.

What is the loss mechanism that causes a larger change on increasing pressure compared to decreasing one?

The discrepancy in barocaloric effects on applying and removing pressure only occurs at low and high temperature, due to losses associated with the hysteresis of the first-order PC-OC transition.

However, our barocaloric effects are reversible at intermediate temperatures, but this was not explained properly in our previous submission. We have therefore changed:

“Discrepancies on applying and removing pressure appear in the magnitude of $\Delta S(T,p)$ [Figure 3(c)] both near and below $T_0(p=0) \sim 314$ K, evidencing irreversibility. By contrast, $\Delta S(T,p)$ is reversible a few kelvin above $T_0(p=0)$.”

to:

“Discrepancies in the magnitude of $\Delta S(T,p)$ on applying and removing pressure [Figure 3(c)] are absent in the range ~314-342 K, evidencing reversibility.”

Further I have some remarks on the current manuscript the figures seem to have been renumbered without change in the text.

We apologise for this error, which is now corrected.

Figure 4a should be presented in double logarithmic scale to give any meaning to the green and orange datapoints.

We failed to explain that the green rectangle represents an envelope containing data points, rather than any data points themselves. The relevant part of the Fig. 4a caption is therefore corrected to read:

“For comparison, the green envelope represents state-of-the-art BC materials (Table 1) that operate near room temperature, and the orange symbol represents the standard commercial fluid refrigerant¹⁸ R134a for which operating pressures are ~0.001 GPa.”

A double logarithmic scale would certainly be a good idea in order to resolve the details that appear in Table 1, but our use of a linear scale for the vertical entropy axis makes it clear that NPG outperforms other barocaloric materials, while our use of a linear scale for the horizontal pressure axis makes it clear that large pressures are required.

Note that we have modified the shape of the green envelope in Figure 4(a) in order to represent more precisely the behaviour of existing barocaloric materials at low pressures, where $\Delta S \rightarrow 0$ when $p \rightarrow 0$.

Reference 51 could not be found.

We apologise for this, and we have corrected the details of reference 51.

Response to Referee #1

The authors have clearly found and presented in a scientifically sound manner their discovery of a colossal barocaloric effect in a plastic crystal, providing a pioneering outlook to refrigeration alternative to conventional gas-compression methods.

We thank the Referee for writing that we “have clearly found and presented in a scientifically sound manner (our) discovery of a colossal barocaloric effect in a plastic crystal”, and that our paper represents “a pioneering outlook to refrigeration alternative to conventional gas-compression methods”.

There is certainly much room to fully understanding the mechanism behind the effect. However, the authors had provided sufficient explanations already at the beginning, and have clarified the doubts about these explanations that especially referee 3 had raised. Referee 4's concerns have likely been adequately addressed. There is no doubt about the results and the explanations cannot go beyond a certain point at the initial state of such a finding.

We thank the Referee for his/her endorsement with respect to our responses to Referee 3 and Referee 4, and for writing that we “had provided sufficient explanations already at the beginning, and have clarified the doubts about these explanations”.

At this state, all issues have been addressed adequately so that I find the paper perfectly worthy of publication in its present form.

We thank the Referee for writing that “all issues have been addressed adequately” and for recommending the manuscript for publication for the third time.